# The release of inhibition model reproduces kinetics and plasticity of neurotransmitter release in central synapses

Christopher A. Norman[1,2,3], Shyam S. Krishnakumar [1,4✉], Yulia Timofeeva [1,2✉] & Kirill E. Volynski [1,5✉]

Calcium-evoked release of neurotransmitters from synaptic vesicles (SVs) is catalysed by SNARE proteins. The predominant view is that, at rest, complete assembly of SNARE complexes is inhibited ('clamped') by synaptotagmin and complexin molecules. Calcium binding by synaptotagmins releases this fusion clamp and triggers fast SV exocytosis. However, this model has not been quantitatively tested over physiological timescales. Here we describe an experimentally constrained computational modelling framework to quantitatively assess how the molecular architecture of the fusion clamp affects SV exocytosis. Our results argue that the 'release-of-inhibition' model can indeed account for fast calcium-activated SV fusion, and that dual binding of synaptotagmin-1 and synaptotagmin-7 to the same SNARE complex enables synergistic regulation of the kinetics and plasticity of neurotransmitter release. The developed framework provides a powerful and adaptable tool to link the molecular biochemistry of presynaptic proteins to physiological data and efficiently test the plausibility of calcium-activated neurotransmitter release models.

[1] University College London Institute of Neurology, University College London, London WC1N 3BG, UK. [2] Department of Computer Science, University of Warwick, Coventry CV4 7AL, UK. [3] Mathematics for Real-World Systems Centre for Doctoral Training, University of Warwick, Coventry CV4 7AL, UK. [4] Department of Neurology, Yale Nanobiology Institute, Yale University School of Medicine, New Haven, CT 06510, USA. [5] Department of Cell Biology, Yale University School of Medicine, New Haven, CT 06510, USA. ✉email: shyam.krishnakumar@yale.edu; y.timofeeva@warwick.ac.uk; k.volynski@ucl.ac.uk

Synaptic release of neurotransmitters forms the basis of information transfer in the brain. It is well established that synaptic vesicle (SV) fusion with the plasma membrane is mediated by SNARE (soluble N-ethylmaleimide–sensitive factor attachment protein receptor) proteins, namely VAMP2 on the SV (v-SNARE) and syntaxin1 and SNAP25 on the plasma membrane (t-SNAREs) in most central synapses[1–3]. v- and t-SNAREs can constitutively assemble (or 'zipper') into a complex that brings opposing membranes together and provides the energy required for fusion. In addition to synapses, similar SNARE proteins mediate the fusion of virtually all membranous organelles in living cells[4].

A distinct property of SV exocytosis is that it is tightly coupled to neuronal activity and controlled by action potential (AP)-evoked increases in presynaptic [$Ca^{2+}$]. To achieve this, presynaptic terminals maintain a readily releasable pool (RRP) of vesicles that are docked at the presynaptic active zone (AZ). When an AP reaches the presynaptic terminal, it depolarises the presynaptic membrane and transiently activates voltage-gated $Ca^{2+}$ channels (VGCCs) located in the AZ, resulting in the formation of local $Ca^{2+}$ nano/microdomains near RRP vesicles ([$Ca^{2+}$]$_{local}$ ~ 10–100 μM). $Ca^{2+}$ ions activate the fast, low-affinity vesicular $Ca^{2+}$ release sensor synaptotagmin 1 (Syt1, or its closely related isoforms Syt2 and Syt9 in different types of synapses), which triggers SV exocytosis and synchronous neurotransmitter release on a millisecond timescale[3,5]. VGCCs close within several milliseconds after an AP, resulting in the collapse of $Ca^{2+}$ nano/microdomains. However, the presynaptic $Ca^{2+}$ level remains elevated in the low micromolar range for tens to hundreds of milliseconds. This long-lasting increase in residual [$Ca^{2+}$] ([$Ca^{2+}$]$_{residual}$), which is especially prominent during bursts of neuronal activity, triggers delayed asynchronous neurotransmitter release and also contributes to the facilitation of synchronous release upon arrival of another AP. This short-term plasticity of vesicular release allows presynaptic terminals to process the neuronal spiking code and provides a basis for synaptic computation and selective information transfer in the brain[6,7]. Asynchronous release and synaptic facilitation are, in large part, mediated by the presynaptic membrane-associated high-affinity $Ca^{2+}$ release sensor synaptotagmin 7 (Syt7), which can be activated by [$Ca^{2+}$]$_{residual}$[8–11]. Thus, it is emerging that the synaptic release of neurotransmitters is synergistically regulated by low- and high-affinity synaptotagmins acting on the same pool of vesicles or even on the same SNARE complex. How this occurs in molecular terms remains poorly understood.

The current prevailing view is that each RRP vesicle contains several partially assembled SNARE complexes ('SNAREpins') that are arrested ('clamped') in this state by synaptotagmins and the soluble presynaptic protein complexin. The SNAREpins are thought to be released by $Ca^{2+}$-activation of synaptotagmin molecules and act cooperatively to drive rapid SV exocytosis leading to neurotransmitter release[12–14]. However, it has not been quantitatively tested whether this 'release of inhibition' model can adequately describe the millisecond kinetics of synchronous neurotransmitter release or if additional mechanisms, such as membrane bending and/or membrane bridging by Syt1[15–21], are also critical.

Recently, considerable progress has been made in understanding the structural and functional organisation of the SNARE pre-fusion complex[21–25]. For example, it has been shown that a given SNAREpin can simultaneously bind two Syt1 molecules, one at the 'primary' interface, independently of complexin, and another at the 'tripartite' interface, in conjunction with complexin (Fig. 1a)[21]. Interestingly, the structural analysis suggests that the primary site might be accessible to only fast, low-affinity $Ca^{2+}$ sensors (Syt1, Syt2 and Syt9), whilst the tripartite site appears to be universally accessible to all synaptotagmin isoforms, including Syt7[21]. Dual synaptotagmin binding at primary and tripartite interfaces has the potential to explain, in molecular terms, how different synaptotagmin isoforms cooperatively regulate neurotransmitter release and short-term plasticity[10–13]. However, this hypothesis remains to be tested.

One difficulty in addressing these questions is that measuring the spatio-temporal dynamics of $Ca^{2+}$ at the AZ is challenging due to its spatial scale (200–600 nm) and the inherently low signal-to-noise ratio of fluorescent $Ca^{2+}$ indicators when imaging with millisecond resolution. Another obstacle is that, at present, it is not possible to directly track the molecular states of different synaptotagmin isoforms on RRP vesicles. Data-constrained realistic computational models of presynaptic terminals are, therefore, essential tools that can bypass the limitations of experimental approaches. Indeed, we have previously created a set of computational tools to model presynaptic $Ca^{2+}$ dynamics at different types of synapses during physiological patterns of activity[26–30].

Here we describe an experimentally constrained computational modelling framework that allows us to model the activation of Syt1 and Syt7 by physiologically relevant $Ca^{2+}$ transients that occur at the presynaptic AZ and to test how their activation triggers SV exocytosis for different molecular models of the fusion clamp. We find that release of inhibition is sufficient to explain the millisecond kinetics of AP-evoked SV exocytosis. Our results indicate that, irrespective of the triggering $Ca^{2+}$ signal's shape, or the nature of the fusion clamp, the majority of synchronous vesicular fusion occurs when 3 SNAREpins are simultaneously free from inhibition. Furthermore, our simulations reveal that the Syt1/SNARE interaction at the primary site alone can account for the millisecond kinetics of AP-evoked synchronous release and that binding of Syt1 or Syt7 to SNARE complexes at the tripartite interface provides an additional level of regulation of vesicular fusion. In particular, dual Syt1/Syt7 binding to the same SNAREpin can explain the role of Syt7 in the regulation of short-term synaptic plasticity and the kinetics of vesicular release.

## Results

**Computational implementation of the release of inhibition model.** We assumed that each RRP vesicle was associated with several partially assembled SNAREpins that were clamped in this state by Syt1 and Syt7 isoforms along with complexin (Fig. 1a). We considered three synaptotagmin clamp architectures based on the available structural and functional data. In all cases, the primary interface was occupied by Syt1. Indeed, the primary interface appears to be selective for Syt1 and its similar isoforms, Syt2 and Syt9[21]. It has also been shown that Syt1 can simultaneously interact with the lipid bilayer via PIP2 interaction and with SNAREs via the primary interface[24]. It is thus likely that Syt1 binding at the primary interface occurs at an early stage of vesicle docking, preceding the SNARE assembly process. In contrast, the tripartite interface is generated only when SNAREs are partially zippered and can bind complexin. Furthermore, the tripartite interface binding motif is present in both Syt1 and Syt7[21]. Based on these data, we considered three distinct synaptotagmin fusion clamp architectures. In all cases, the primary interface was occupied by Syt1, whilst the tripartite interface was either unoccupied (single Syt1 clamp at the primary interface, Syt1$^P$) or occupied (dual clamp) either by Syt1 (Syt1$^P$/Syt1$^T$) or by Syt7 (Syt1$^P$/Syt7$^T$) (Fig. 1a). To model these three limiting cases, we assumed that all SNAREpins on a given RRP vesicle share the same clamp architecture.

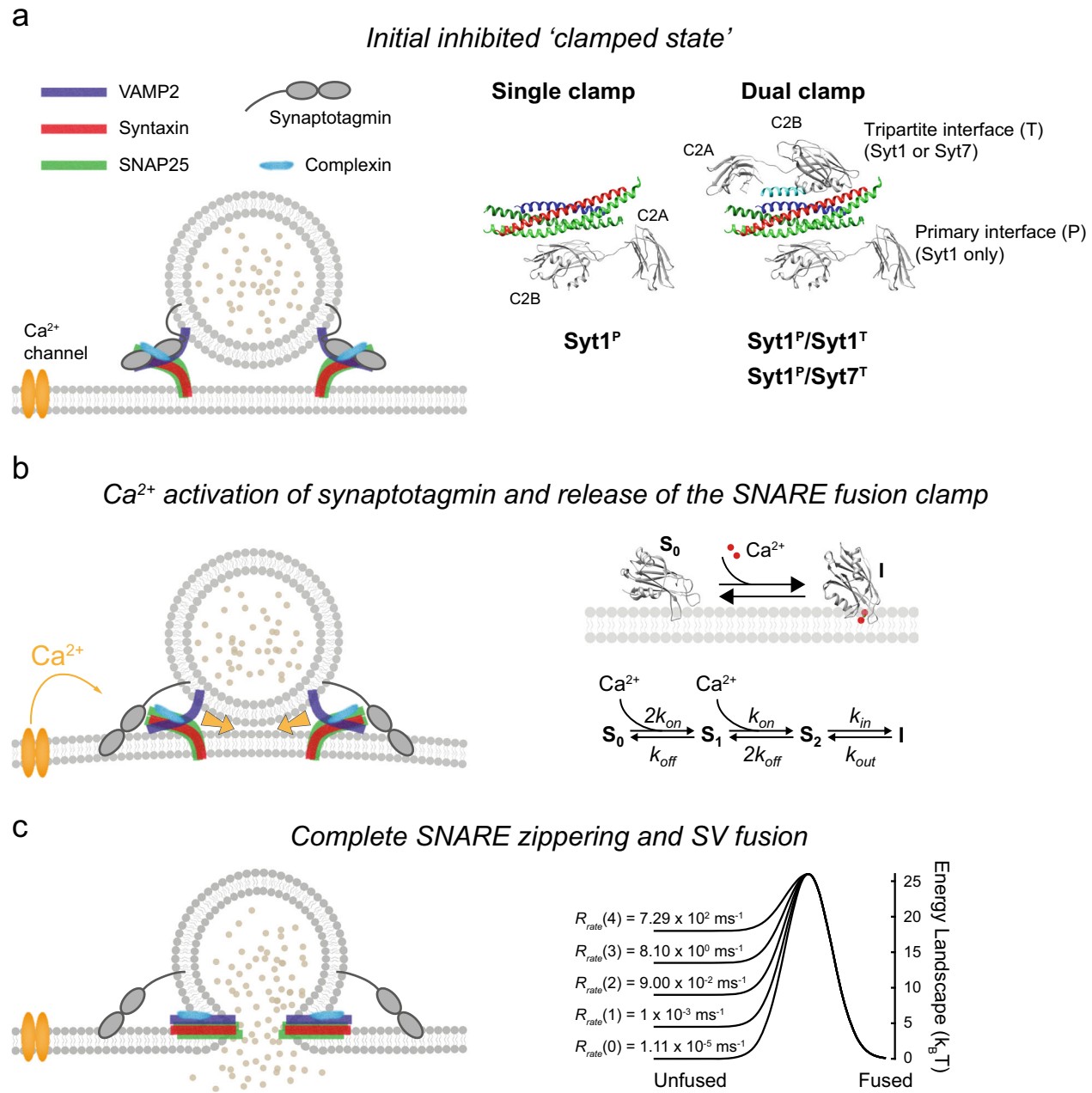

**Fig. 1 Computational implementation of the release of inhibition model. a** At rest, the full zippering of SNAREs on RRP vesicles is inhibited ('clamped') by the binding of synaptotagmin and complexin molecules. Based on structural data, three synaptotagmin/SNARE clamp architectures were considered in the model (right). In all cases, Syt1 occupies the primary interface. The tripartite interface is either unoccupied (single clamp, Syt1$^P$) or occupied (dual clamp) by Syt1 (Syt1$^P$/Syt1$^T$) or Syt7 (Syt1$^P$/Syt7$^T$). Crystal structure (PDB ID: 5W5C)[21]. **b** The binding of two Ca$^{2+}$ ions to a synaptotagmin C2 domain leads to its subsequent membrane insertion, described by the Markov kinetic scheme on the right, and release of its SNARE fusion clamp, allowing full zippering of SNAREs which provides energy for membrane fusion. **c** The rate of SV fusion is determined by the number of free SNARE complexes. These reduce the effective membrane fusion energy barrier, illustrated here as a Gaussian landscape (right). Each SNAREpin was assumed to independently contribute to the lowering of the membrane fusion barrier, which is spontaneously overcome at a rate given by the Arrhenius equation (see "Methods"). Note that only two out of six SNARE complexes are shown in the cartoons on the left that represent a vertical cross-section of the SV.

Biochemical and physiological analyses have shown that a small number of SNAREs are sufficient to achieve fast, Ca$^{2+}$ synchronised neurotransmitter release[31–33]. Furthermore, recent cryo-electron tomography analysis in cultured hippocampal synapses demonstrated a circular symmetric arrangement of six protein densities at the interface between docked SVs and the presynaptic membrane, each possibly corresponding to a single SNARE-associated exocytic module[34]. Therefore, as a default setting, we assumed that each RRP vesicle was associated with six SNAREpins.

It is well established that Ca$^{2+}$ binding leads to rapid insertion of the aliphatic loops of synaptotagmin C2 domains into the membrane and that this step is critical for triggering neurotransmitter release[35,36]. Indeed, structural and biochemical analyses have indicated that a Ca$^{2+}$-triggered reorientation of Syt1 C2 domains displaces Syt1 from the primary SNARE interface[24,37]. Therefore, we assumed that Ca$^{2+}$ binding and subsequent membrane loop insertion of synaptotagmin C2 domains induces the removal of the fusion clamp, i.e. 'release

of inhibition' (Fig. 1b). Based on the previously established critical roles of the Syt1 C2B domain[38,39] and the Syt7 C2A domain[8,9], for simplicity, we only considered activation of these domains in our model. The C2 domains of Syt1 and Syt7 associated with RRP vesicles are likely to be in close proximity to the membrane[24,40,41]. Hence, we modelled $Ca^{2+}$-triggered loop insertion as a first-order reaction described by membrane insertion ($k_{in}$) and dissociation ($k_{out}$) rates. Combined with the two-site protein-ligand binding model described by $Ca^{2+}$ binding ($k_{on}$) and unbinding ($k_{off}$) rates, the Markov model of Syt1 and Syt7 C2 domain dynamics is described by the kinetic scheme in Fig. 1b. This model assumes that $Ca^{2+}$ is not able to dissociate from the C2 domain while it is membrane-inserted, and that reversal of membrane insertion leads to immediate restoration of the SNARE fusion clamp.

Finally, we assumed that the repulsive forces between an SV and the plasma membrane constitute a potential energy barrier, which is lowered by the independent energetic contributions of individual assembled SNARE complexes. SV fusion was triggered when the barrier was overcome by thermal fluctuations at a rate given by the Arrhenius equation (Fig. 1c)[42,43]. The parameters of the complete Markov model were constrained using the available biochemical and structural data[13,21,24,40,41,44–48] (see Methods). The fusion dynamics of RRP vesicles in response to diverse $[Ca^{2+}]$ transients were computed using stochastic Monte Carlo simulations using the Gillespie algorithm[49], as detailed in the Methods section.

**The release of inhibition model describes the kinetics of vesicular release at the calyx of Held giant synapse.** The $Ca^{2+}$-activation of SV fusion has been quantitatively described in the calyx of Held—a giant synapse in the auditory brainstem. In these experiments, flash photolysis was used to generate spatially uniform, step-like increases of $[Ca^{2+}]$ within the calyx and the $Ca^{2+}$-evoked vesicular release was monitored using post-synaptic patch-clamp recordings[50,51]. The measured kinetics and the $[Ca^{2+}]$ dependency of glutamate release at this synapse have been described by several empirical mathematical models[50–53]. Among these, the six-state allosteric model proposed by Lou et al. (Supplementary Fig. 1) was shown to closely approximate the kinetics of $Ca^{2+}$-triggered vesicular release in the calyx of Held over a wide range of $[Ca^{2+}]$[52]. Furthermore, this empirical model has been widely used to describe the efficacy and plasticity of $Ca^{2+}$-evoked vesicular release in many other functionally and structurally distinct synapses, including GABAergic terminals of hippocampal parvalbumin-containing basket cells[54], glutamatergic terminals of hippocampal granular[28,55] and CA3[30] pyramidal neurons. Therefore, we used the allosteric model as a benchmark to compare the results of our simulations to the kinetics and plasticity of vesicular release observed in different types of central synapses.

In line with the $Ca^{2+}$ uncaging experiments, we simulated vesicular release in response to $[Ca^{2+}]$ steps in the range of 1–32 μM for the three limiting cases of the clamp architecture (Fig. 1a). In all three cases, the peak release rates exhibited a power law dependency on the $[Ca^{2+}]$ step, with exponents between 2.7 and 4.3 (Fig. 2a, b). In agreement with the experimental data recorded at the calyx of Held described by the benchmark allosteric model, sub-millisecond fusion rates predicted by the release of inhibition models were apparent for $[Ca^{2+}]$ steps above 4 or 8 μM, depending on the clamp architecture. The release rate was greatest in the case of a single Syt1 clamp at the primary interface. The introduction of an additional Syt1 or Syt7 clamp at the tripartite interface reduced the release rate but enhanced $Ca^{2+}$ cooperativity. The predictions

of the allosteric model lie between these limiting clamping cases, demonstrating that the release of inhibition model can indeed explain the experimentally observed kinetics of vesicular release. The model output further suggests that the occupancy of the tripartite interface by different synaptotagmin isoforms could provide an efficient mechanism for the dynamic regulation of $Ca^{2+}$-triggered vesicular release.

To estimate how many SNARE complexes are needed to drive fast synchronous vesicular fusion, we monitored the number of unclamped SNAREpins associated with each vesicle prior to fusion. We found that for all clamp architectures, fast fusion (peak release rate above $10^{-2}$ ms$^{-1}$) required at least three uninhibited SNAREpins. This value is consistent with previous experimental and modelling estimates for the number of SNAREs required to mediate synchronous release of neurotransmitters[31,32,43]. Indeed, the apparent rate of synchronous vesicle fusion in our modelling framework is limited by the time taken for three SNAREpins to be released from the fusion clamp, which depends on the $[Ca^{2+}]$ increment and the clamp architecture (Fig. 2c). In the case of three unclamped SNAREpins the vesicular fusion rate predicted in our model by the Arrhenius equation is 8.1 ms$^{-1}$ (Fig. 1c). This value is orders of magnitude greater than the upper limit for how quickly the synaptotagmin clamp can be restored, which is limited by the rate of C2 domain membrane dissociation ($k_{out} = 0.67$ ms$^{-1}$ for Syt1 and $k_{out} = 0.02$ ms$^{-1}$ for Syt7, see Methods). This implies that once three out of six SNAREpins are simultaneously unclamped, vesicle fusion is practically inevitable. In line with this prediction, the peak release rate was directly proportional to the fraction of vesicles with three uninhibited SNAREpins, irrespective of the synaptotagmin clamp architecture (Fig. 2d).

As the exact number of SNARE complexes in each RRP vesicle is still undetermined, we explored the effect of varying the number of SNAREpins per vesicle (specifically four, six and eight) (Supplementary Fig. 2). We found that, for all clamp architectures, vesicular fusion took place when at least three SNAREpins were free from the fusion clamp, regardless of the total number of SNAREpins on a given vesicle. However, we observed a correlation between the peak fusion rate and the number of available SNAREpins. For every $[Ca^{2+}]$ tested, the peak release rate was at its highest with eight SNAREpins per vesicle and at its lowest with four. This pattern could be explained by the fact that the state with three unclamped SNAREpins was reached more quickly when there was a greater total number of SNAREpins on the vesicle.

**The release of inhibition model recapitulates synchronous release in response to AP-evoked $[Ca^{2+}]$ transients at the AZ.** We next examined whether the release of inhibition model could replicate the vesicular release dynamics observed at presynaptic terminals in response to AP stimulation using the default models described in Fig. 1. AP-evoked $Ca^{2+}$ dynamics at vesicular release sites have not been directly measured, largely because of the small size of the AZ and the high speed of $Ca^{2+}$ kinetics. Therefore, we and others have developed three-dimensional, experimentally constrained models of $Ca^{2+}$ influx, diffusion, buffering and extrusion, which allow one to approximate the $[Ca^{2+}]$ transients near release-ready vesicles at different types of synapses[26–30,54,56]. A critical parameter that determines $[Ca^{2+}]_{local}$ at a given release site is its distance to the nearest VGCC cluster, i.e. the coupling distance, $d$. As in our previous work[29], we considered a simplified model of a small excitatory presynaptic terminal that contains a single AZ with a cluster of VGCCs at the centre and computed AP-evoked $[Ca^{2+}]_{local}$ for coupling distances ranging between 30–80 nm, which is characteristic for small central glutamatergic

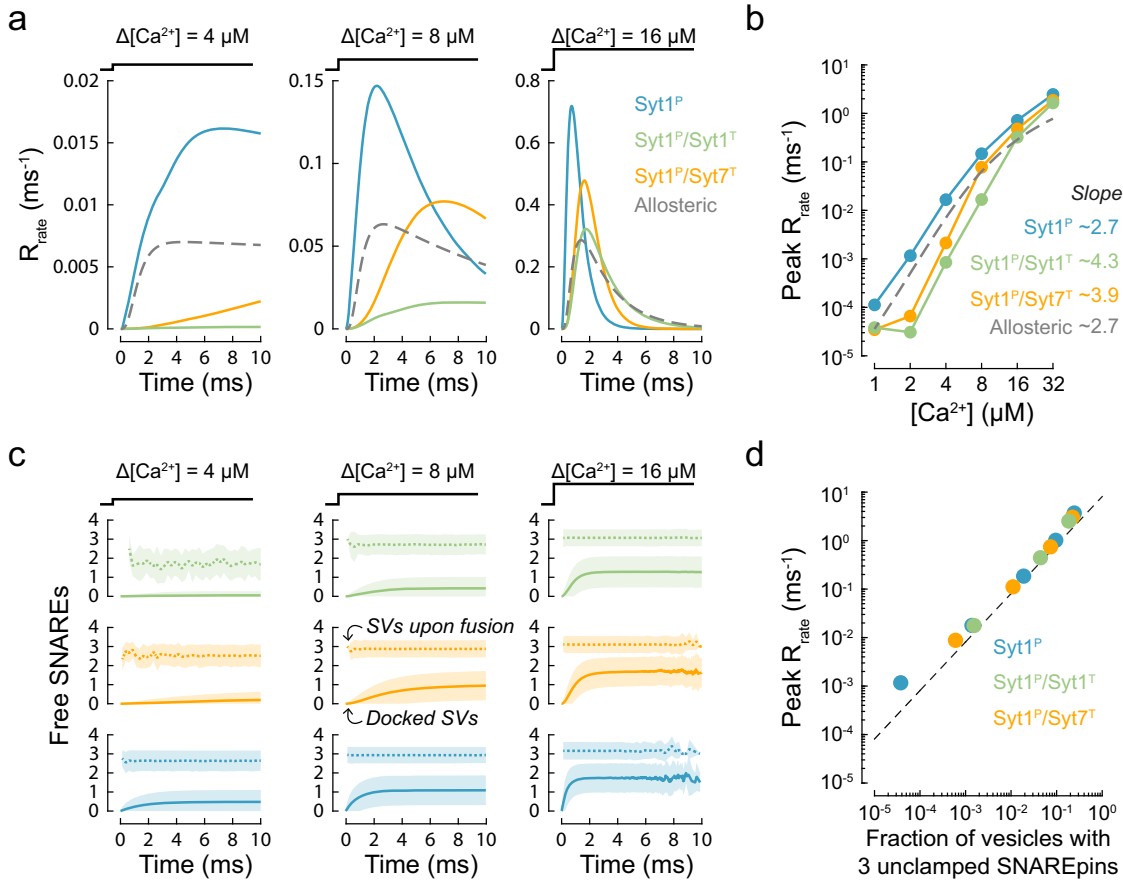

**Fig. 2 Release of inhibition model describes the kinetics of vesicular release in the calyx of Held. a** Time-course of vesicular release rate simulated in response to 4, 8, and 16 μM [Ca²⁺] steps for the single and dual synaptotagmin/SNARE clamp architectures considered in the model (solid coloured traces) and for the benchmark allosteric model (dashed grey trace) that describes vesicular release kinetics recorded in the calyx of Held[52]. **b** Dependency of the peak release rate (achieved within 10 ms) on the amplitude of a [Ca²⁺] step. The numbers indicate the slopes corresponding to the exponent of the power relationship between peak vesicular release rate and [Ca²⁺] in the range of 4–16 μM. **c** Time evolution of the mean number of unclamped SNAREpins ('Free SNAREs') on all docked SVs (solid lines) and on SVs at the instance of fusion (dotted lines) in response to 4, 8, and 16 μM [Ca²⁺] steps. Shaded areas indicate 1 standard deviation on each side of the mean. Each time point includes data from a 0.25 ms bin. The colour code is the same as in (**a**). **d** The relationship between peak release rate and the fraction of SVs that have three unclamped SNARE complexes at the instance of fusion. The dotted line represents an asymptote for the case when fusion may only occur for vesicles that have exactly 3 unclamped SNARE complexes (i.e. the product of the fraction of vesicles with 3 unclamped SNAREs and the Arrhenius rate $R_{rate}(3) = 8.1$ ms⁻¹, see Fig. 1c and Methods). Data points represent mean values taken over a 0.25 ms bin centred on the time of peak release rate. For each [Ca²⁺] step and fusion clamp architecture, at least $N = 500{,}000$ stochastic simulations were performed with at least 2000 vesicular fusion events recorded during the first 10 ms time window. This restricted the normalised root mean squared error in stochastic estimates of the kinetics of SV fusion to less than 1% (see Supplementary Fig. 5).

synapses (Fig. 3a and Methods). We then extracted [Ca²⁺]_local at different coupling distances and used these as model inputs to simulate vesicular fusion (Fig. 3b, c).

In line with the analysis of step [Ca²⁺] increments (Fig. 2a, b), the model-predicted vesicular release kinetics in response to AP-evoked Ca²⁺ influx depended on the architecture of the synaptotagmin clamp (Fig. 3). For all coupling distances tested, the vesicular release probability was highest when the tripartite interface was unoccupied (single Syt1ᴾ clamp). Adding either a second Syt1 or Syt7 clamp at the tripartite interface reduced the vesicular release by similar amounts. The amplitude of AP-evoked [Ca²⁺] transients decreased with increasing coupling distance, which resulted in a corresponding reduction of both the peak release rate and the overall release probability (Fig. 3c). The decrease was steeper when the tripartite site was occupied by either Syt1 or Syt7 due to increased Ca²⁺ cooperativity. Notably, the predictions of the benchmark allosteric model, which describes the experimentally observed vesicular release properties in small glutamatergic synapses, were within the range of the

predictions of the three limiting cases of clamping architectures. These results further demonstrate that the release of inhibition model can reproduce the [Ca²⁺] dependency and fast kinetics of AP-evoked synchronous vesicular release in small excitatory synapses.

**The release of the inhibition model reproduces Syt7-dependent short-term facilitation.** Short-term plasticity of synaptic neurotransmitter release is commonly assessed by measuring vesicular release in response to pairs of APs. We therefore tested how the molecular architecture of the synaptotagmin clamp shapes vesicular release in response to paired-pulse stimulation (Fig. 4). Using the VCell model described in Fig. 3, we computed [Ca²⁺] responses to different inter-stimulus intervals (10–500 ms) and coupling distances (30–80 nm). We next simulated vesicular release for different synaptotagmin clamp architectures in response to the obtained Ca²⁺ dynamics and calculated the paired-pulse ratio PPR $= p_v(2)/p_v(1)$ (where $p_v(1)$ and $p_v(2)$ are vesicular release probabilities at the 1st and the 2nd AP,

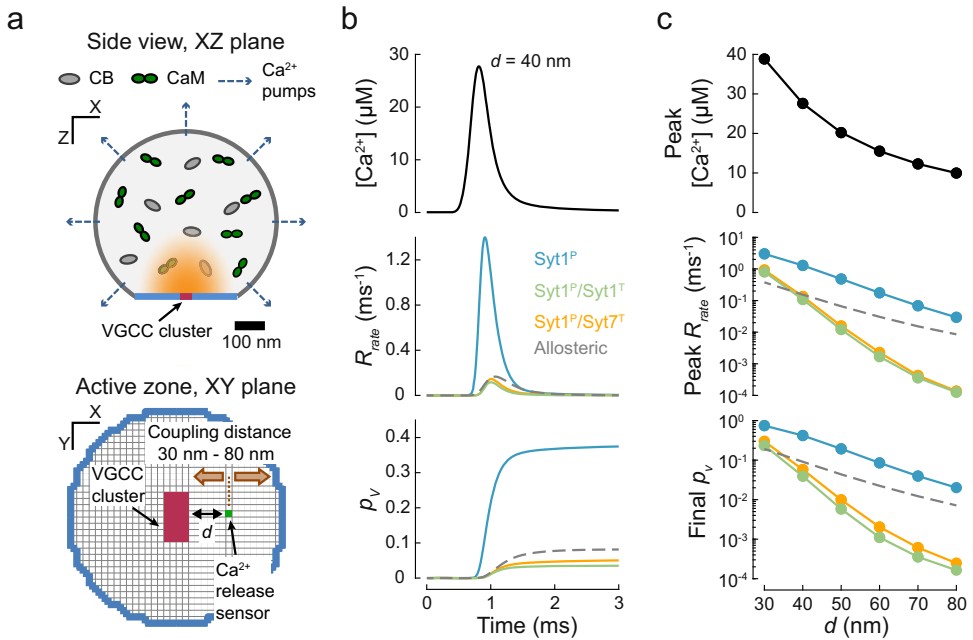

**Fig. 3 The release of inhibition model recapitulates synchronous release in response to AP-evoked [Ca$^{2+}$] transients in the presynaptic AZ. a** A cartoon illustrating the model of AP-evoked presynaptic Ca$^{2+}$ dynamics. The presynaptic terminal was modelled as a truncated sphere with a diameter of 0.6 μm with a single AZ containing a 40 nm × 80 nm rectangular cluster of VGCCs. Intraterminal Ca$^{2+}$ dynamics were subject to diffusion, buffering (by calbindin, CB; calmodulin, CaM; and ATP) and extrusion via ion pumps. Illustrations of the volume cross-section (upper) and the AZ plane (lower) are adapted from our previous work[29]. **b** Time course of AP-evoked Ca$^{2+}$(t) at an AZ point 40 nm from the VGCC cluster (top) and the corresponding vesicular release rates (middle) and the cumulative vesicular release probability, $p_v$ (bottom) for the three clamp architectures (solid coloured lines) and the benchmark allosteric model (dashed grey line). **c** Dependences of peak [Ca$^{2+}$] (top), peak vesicular release rate (middle) and 'final' $p_v$ calculated 2 ms after an AP (bottom) on the coupling distance $d$ between the docked vesicle and the VGCC cluster. Colour codes as in (**b**). Data points represent mean values. For each [Ca$^{2+}$](t) transient and fusion clamp architecture, at least $N = 500{,}000$ stochastic simulations were performed with at least 2000 vesicular fusion events recorded during the first 3 ms time window.

respectively). To specifically track changes in the release probability of a given RRP vesicle during paired-pulse stimulation, we did not model SV replenishment in this set of simulations.

In the case of the benchmark allosteric model, vesicular release was similar in response to the 1st and 2nd AP (Fig. 4a–c). This was expected since [Ca$^{2+}$] transients at vesicular release sites were comparable for the 1st and 2nd AP (Fig. 4a). In the case of the single Syt1 clamp at the primary interface (Syt1$^P$), we observed depression of vesicular release in response to the 2nd AP, which was most prominent at short coupling distances where [Ca$^{2+}$]$_{local}$ transients had the greatest amplitude. The depression can be explained by the depletion of the RRP vesicles due to the absence of vesicle replenishment in these simulations and the relatively high release probability at the 1st AP (Fig. 3c and Fig. 4a). In contrast, in the case of the dual Syt1 clamp (Syt1$^P$/Syt1$^T$) the release probability at both the 1st and the 2nd APs was low, and the PPR was close to 1 for the whole range of coupling distances and inter-stimulus intervals tested. Strikingly, the inclusion of Syt7 at the tripartite interface (Syt1$^P$/Syt7$^T$) led to a facilitation of vesicular release at the 2nd AP (Fig. 4a–c). The increase of PPR was most noticeable (2- to 3-fold) at shorter inter-stimulus intervals (10–50 ms) and longer coupling distances (> 40 nm).

The observed facilitation can be explained by the slower membrane dissociation kinetics of Syt7 relative to Syt1[45]. The 1st AP induces the insertion of Syt1 and Syt7 C2 domains into the membrane, which leads to the release of the fusion clamp at a fraction of the total SNAREpins. If the fusion of a given vesicle was not successfully induced during the 1st AP, then the clamp on its SNAREpins would be restored more slowly by Syt7 than by Syt1, because Syt7 stays in the membrane longer ($k_{out} = 0.02$ ms$^{-1}$ for Syt7 and $0.67$ ms$^{-1}$ for Syt1). This means that at

the time of arrival of the 2nd AP these unfused vesicles are expected to have up to 40% of Syt7 clamps already released (Fig. 4d), conferring an increase in the probability of vesicle fusion, $p_v$(2).

The observed short-term facilitation mediated by Syt7 diminished as the inter-stimulus interval increased, due to progressive restoration of the Syt7 clamp, and disappeared within 500 ms, consistent with experimental data[11,57]. Indeed, the dependency of PPR on the inter-stimulus interval for a given coupling distance aligned well with that of the average number of free Syt7 clamps immediately before onset of the 2nd AP (Fig. 4c, d). In contrast, the dependency of Syt7-mediated short-term facilitation on coupling distance was non-monotonic, with PPR reaching maximal values at coupling distances between 50–60 nm (Fig. 4b). This was due to the competing effects of increased vesicle depletion after the 1st AP at short coupling distances versus decreased removal of Syt7 clamps on unfused vesicles at longer coupling distances.

**Mixed single and dual fusion clamp models can recapitulate vesicular release properties at plastic synapses.** We next assessed whether the release of inhibition model could explain the complex patterns of vesicular release observed in response to bursts of neuronal activity. To test this, we chose the mossy fibre bouton (MFB)—CA3 pyramidal neuron synapse in the hippocampus as a model environment. This synapse, which is also called a 'detonator synapse', has a very low initial release probability ($p_v$ for individual RRP vesicles is in the range of 0.01–0.03) and shows strong short-term facilitation of synchronous release (up to 10-fold) and prominent asynchronous release after high frequency bursts of activity[55,58]. Short-term facilitation in MFBs is mediated by at least

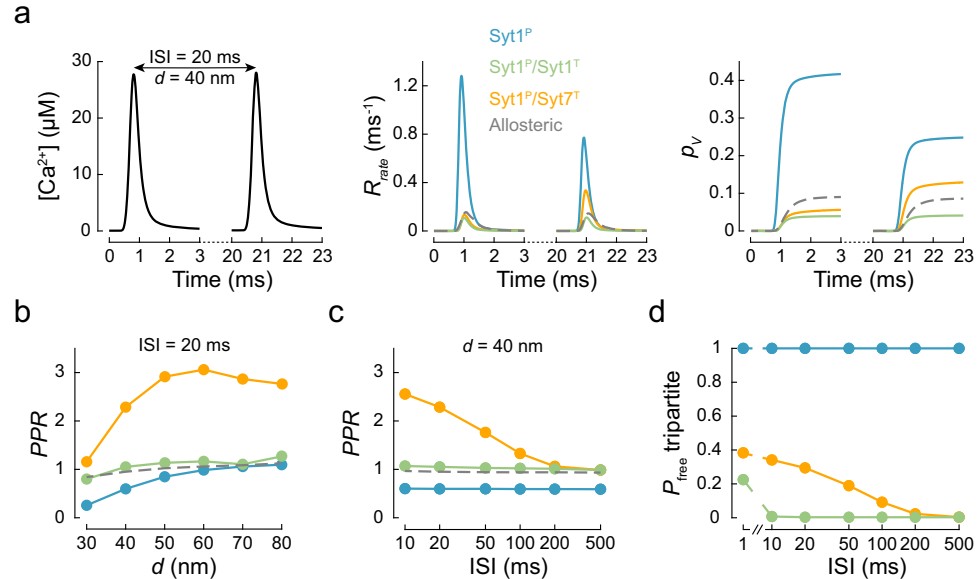

**Fig. 4 The release of inhibition model reproduces Syt7-dependent short-term facilitation. a** Modelling of vesicular release in response to paired-pulse stimulation. Left, VCell-computed $Ca^{2+}$ dynamics at a vesicular release site located $d = 40$ nm from the VGCC cluster (see Fig. 3a) in response to a pair of APs separated by an inter-stimulus interval (ISI) of 20 ms. Time course of the corresponding vesicular release rates (middle) and the cumulative vesicular release probability, $p_v$ (right) for the three fusion clamp architectures (solid coloured lines) and the benchmark allosteric model (dashed grey line).
**b, c** Dependency of the paired-pulse ratio (PPR = $p_v(2)/p_v(1)$, where $p_v(1)$ and $p_v(2)$ are the vesicular release probabilities at the 1st and the 2nd AP respectively), on the coupling distance (in **b**) and the inter-stimulus interval (ISI) (in **c**) for different clamp architectures. Note that facilitation of vesicular release was only observed when Syt7 was present (Syt1$^P$/Syt7$^T$). Colour codes as in (**a**). **d** Probability that the tripartite interface remains unoccupied ($P_{free}$ tripartite) at the time of arrival of the 2nd AP. Due to the faster membrane dissociation rate of Syt1 ($k_{out} = 0.67$ ms$^{-1}$), the model predicts the restoration of the Syt1 clamp within 10 ms after the 1st AP. In contrast, Syt7 exhibits slower membrane dissociation ($k_{out} = 0.02$ ms$^{-1}$), which leads to a delayed restoration of the fusion clamp and results in the facilitation of vesicular release at the 2nd AP. Colour codes as in (**a**). Data points represent mean values. For each [$Ca^{2+}$]($t$) transient and fusion clamp architecture, at least $N = 500,000$ stochastic simulations were performed with at least 2000 vesicular fusion events recorded for each AP.

two different mechanisms: (i) progressive increase of peak [$Ca^{2+}$]$_{local}$ at vesicular release sites due to $Ca^{2+}$ buffer saturation and (ii) activation of Syt7 due to increase of [$Ca^{2+}$]$_{residual}$[8,28].

We previously estimated $Ca^{2+}$ dynamics at MFB release sites during high-frequency bursts of APs using a three-dimensional VCell model[27,28]. Here we used the previously estimated [$Ca^{2+}$]$_{local}$ transient at the AZ in response to a 100 Hz train of 10 APs as a model input and simulated vesicular release responses for the three fusion clamp architectures and the benchmark allosteric model (Fig. 5). In this set of simulations we also implemented vesicle replenishment after a fixed refractory time of 2.5 ms with a rate of $\hat{k}_{rep} = 0.02$ ms$^{-1}$ (as estimated in ref. [28]). Therefore, we used the release efficacy, $n_T$, defined as the expected number of vesicles exocytosed at a single release site, instead of $p_v$, to compare vesicular release among different models.

The release efficacy after the first AP predicted by the benchmark model was in the physiological range, $n_T(1) = 0.026$. The allosteric model also replicated short-term facilitation, with synchronous release at the 10th stimulus approximately 4-fold higher than at the 1st stimulus, $n_T(10)/n_T(1) = 4.1$. This degree of facilitation is somewhat lower than the values which have been observed experimentally: $n_T(10)/n_T(1)$ in the range of 5–10[8,28]. Indeed, by its design, the allosteric model accounts for facilitation triggered by increases in peak [$Ca^{2+}$]$_{local}$ at vesicular release sites (Fig. 5a) but not the effect of Syt7 activation resulting from a build-up of [$Ca^{2+}$]$_{residual}$.

In the case of a single Syt1 clamp (Syt1$^P$) the initial release efficacy was several-fold higher than in the case of the benchmark allosteric model, $n_T(1) = 0.07$, and the model predicted only modest facilitation with $n_T(10)/n_T(1) = 2.5$. The addition of a second Syt1 or Syt7 clamp essentially eliminated vesicular fusion

at the 1st AP with $n_T(1) = 0.0008$ for Syt1$^P$/Syt1$^T$ and 0.0015 for Syt1$^P$/Syt7$^T$. Consequently, both dual clamp models showed very strong facilitation, $n_T(10)/n_T(1) = 20$ for Syt1$^P$/Syt1$^T$ and 100 for Syt1$^P$/Syt7$^T$ (Fig. 5a–c).

Considering that neither the single nor the dual clamp architecture could fully recapitulate all the different facets of vesicular release at MFBs, we tested if a mixture of single and dual synaptotagmin clamps could better describe the physiological data. We considered two models. In both cases, on average, half of the SNAREpins were clamped by Syt1 at the primary interface only, whilst the remaining SNAREpins had a dual clamp arrangement with either Syt1 or Syt7 at the tripartite interface (Mixed Syt1 model and Mixed Syt7 model respectively, Fig. 6a). In comparison to the full dual clamp models considered above, partial removal of the clamp from the tripartite interface increased the initial release efficacy for individual RRP vesicles to the physiological level, $n_T(1) = 0.014$ for the Mixed Syt1 model and 0.015 for the Mixed Syt7 model (Fig. 6b, c). Both mixed models showed prominent short-term facilitation, which was stronger for the Mixed Syt7 model, $n_T(10)/n_T(1) = 10.6$, than the Mixed Syt1 model, $n_T(10)/n_T(1) = 6.1$. This is because the progressive increase of peak [$Ca^{2+}$]$_{local}$ is the sole determinant of facilitation in the case of the Mixed Syt1 model. In contrast, Syt7 is activated by the elevation of [$Ca^{2+}$]$_{residual}$, which leads to progressive removal of Syt7 clamps during the burst in the case of the Mixed Syt7 model (see also Fig. 4d). This result is consistent with the decreased short-term facilitation observed in MFB synapses of Syt7 knockout versus wild-type mice[8].

We also tracked the asynchronous release component triggered by elevated [$Ca^{2+}$]$_{residual}$ after the AP burst (Fig. 5c and Fig. 6d). As expected, we found that the rate of asynchronous release

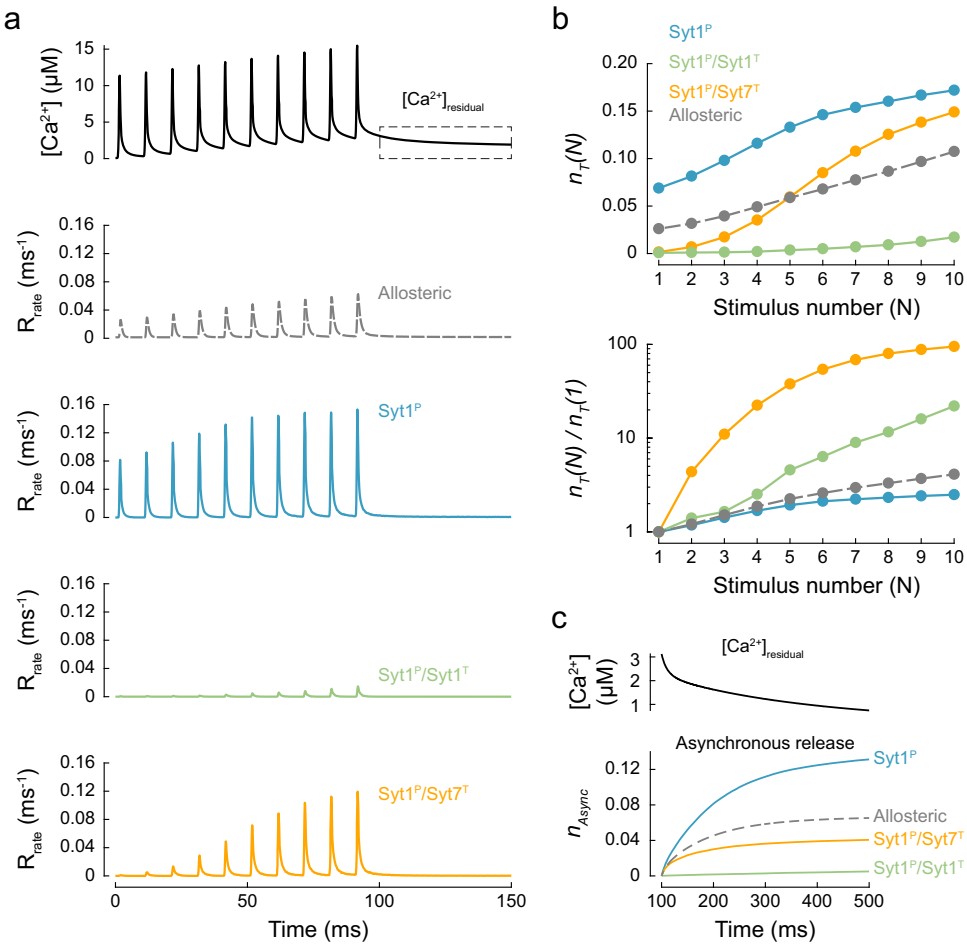

**Fig. 5 Modelling vesicular release for single and dual clamp models in response to bursts of action potentials in MFB terminals. a** Top, VCell-computed $[Ca^{2+}](t)$ transient approximating $Ca^{2+}$ dynamics at vesicular release sites in MFB terminals in response to a $10 \times 100$ Hz AP train (see ref. [28]). Bottom, time course of corresponding vesicular release rates for the benchmark allosteric model and the three fusion clamp architectures. **b** Top, expected numbers of vesicles exocytosed at a single release site, $n_T(N)$, for each AP of the train, $N$. Bottom, paired-pulse ratio: release efficacy at each AP normalised to release efficacy at the first AP, $n_T(N)/n_T(1)$. **c** Top, $[Ca^{2+}]_{residual}$ after the $10 \times 100$ Hz AP train, corresponding to the dashed box in (**a**). Bottom, corresponding cumulative asynchronous releases per release site, $n_{Async}$, for the three fusion clamp architectures and the benchmark allosteric model. Data points represent mean values. For each $[Ca^{2+}](t)$ transient and fusion clamp architecture at least $N = 100,000$ stochastic simulations were performed with at least 6000 vesicular fusion events recorded during the AP train.

depended on the fusion clamp architecture. Asynchronous release was prominent in the single $Syt1^P$, dual $Syt1^P/Syt7^T$, and Mixed Syt1 and Syt7 models, whereas no asynchronous release was observed in the case of the dual $Syt1^P/Syt1^T$ clamp model.

These results further illustrate that the balance between single and dual synaptotagmin clamp arrangements could provide a mechanism for the regulation of both short-term synaptic plasticity and the kinetics of vesicular release.

## Discussion

Our computational analysis shows that the release of inhibition model, i.e. $Ca^{2+}$-triggered removal of the SNARE fusion clamp, can indeed explain the kinetics of evoked neurotransmitter release. Our simulations further demonstrate that fast SV fusion requires simultaneous release of inhibition of at least three SNAREpins. Thus, the kinetics of vesicular fusion depend on how rapidly this state is reached, which is in turn determined by the shape and the amplitude of the $[Ca^{2+}]$ transient at the AZ and by the architecture of the fusion clamp.

Based on the available structural data[21], we considered three possible limiting cases of the fusion clamp architecture: with a single Syt1 clamp at the primary interface ($Syt1^P$) and dual Syt1

and/or Syt7 clamps at primary and tripartite interfaces ($Syt1^P/Syt1^T$ and $Syt1^P/Syt7^T$). Our analysis shows that the release of a single or dual synaptotagmin clamp can account for sub-millisecond kinetics of $Ca^{2+}$-triggered neurotransmitter exocytosis. Furthermore, the dual binding $Syt1^P/Syt7^T$ arrangement also reproduced facilitation of SV exocytosis in response to pairs or burst of APs. This result shows that the release of inhibition model also provides a mechanism by which Syt7 can regulate short-term plasticity.

The functional importance of the primary interface has been well established, both in live synapses and under reconstitution conditions[22,23,33,59]. In contrast, the relevance of the tripartite interface remains unclear because the interaction of SNAREs/synaptotagmins/complexin at this site cannot be measured biochemically and is thus expected to be very weak[21,33,37,60,61]. However, considering the high local concentration of the vesicular release machinery components at RRP vesicles, it is reasonable to expect that the tripartite interface can be at least partially occupied under physiological conditions. In fact, our simulations argue that the weak interaction at this site may play an important functional role and the dynamic occupancy of the tripartite interface by either Syt1 or Syt7 could provide direct

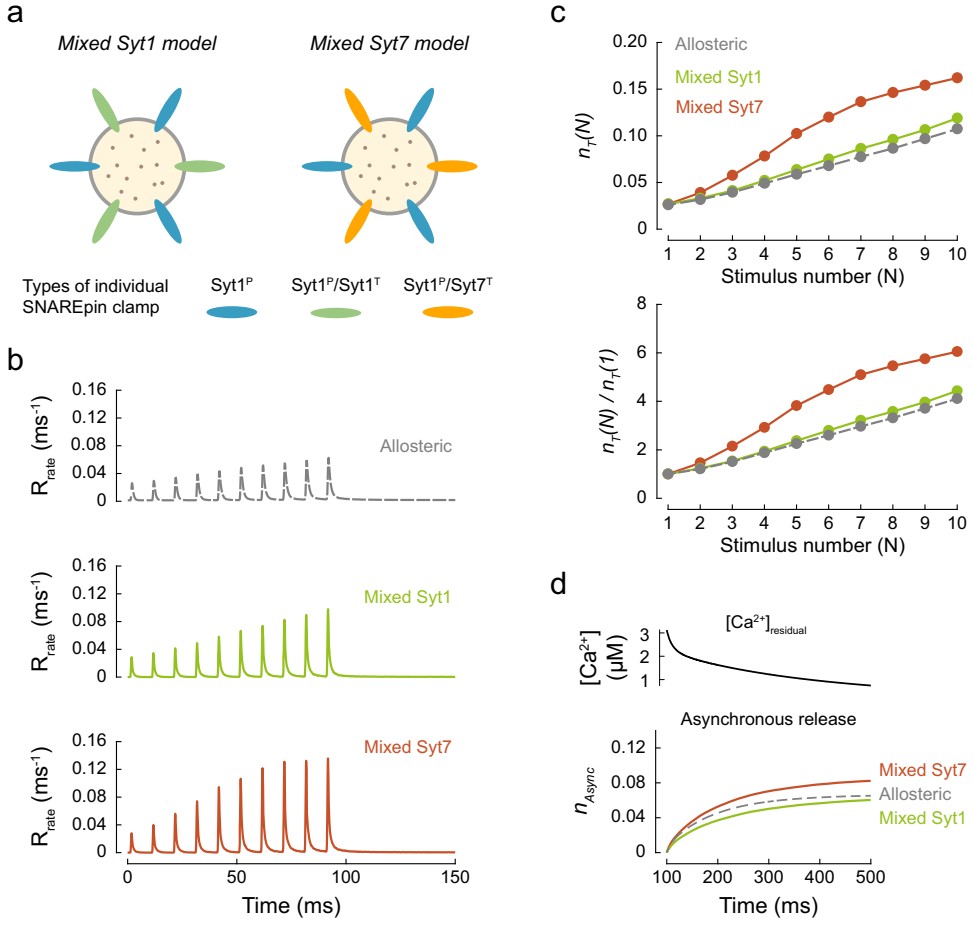

**Fig. 6 Mixed single and dual clamp models can describe the kinetics and plasticity of release in MFBs. a** We considered two partial dual clamp models where on average, 3 out of 6 SNAREpins had a single $Syt1^P$ clamp, and the other 3 had dual $Syt1^P/Syt1^T$ or $Syt1^P/Syt7^T$ clamps. **b** Time course of vesicular release rates for the benchmark allosteric model and the Mixed Syt1 and Mixed Syt7 models in response to the same $[Ca^{2+}](t)$ transient as in Fig. 5a. **c** Top, expected numbers of vesicles exocytosed at a single release site, $n_T(N)$, for each AP of the train, $N$. Bottom, paired-pulse ratio: release efficacy at each AP normalised to release efficacy at the first AP, $n_T(N)/n_T(1)$. **d** Top, $[Ca^{2+}]_{residual}$ after the $10 \times 100$ Hz AP train, corresponding to the dashed box in Fig. 5a. Bottom, corresponding cumulative asynchronous releases per release site, $n_{Async}$, for the two mixed clamp architectures and the benchmark allosteric model. Data points represent mean values. For each $[Ca^{2+}](t)$ transient and fusion clamp architecture, at least $N = 150{,}000$ stochastic simulations were performed with at least 100,000 vesicular fusion events recorded during the AP train.

control of the kinetics and plasticity of neurotransmitter release by changing the strength and $Ca^{2+}$-activation properties of the SV fusion clamp. Indeed, we find that a mixed model combining both single and dual clamp architectures can closely describe the release kinetics and short-term plasticity in response to trains of APs in hippocampal MFB terminals.

While our parsimonious model describes the key molecular elements of $Ca^{2+}$-activation of synaptotagmin and removal of the fusion clamp, it has several simplifications. We assumed that release of the fusion clamp occurs simultaneously with $Ca^{2+}$-triggered membrane insertion of the C2 domains for both Syt1 and Syt7. This is likely the case for Syt1 interaction at the primary interface[24], but the kinetics of synaptotagmin/complexin/SNARE interactions at the tripartite interface are not currently known. Similarly, we assumed that restoration of the clamp occurs instantaneously after reversal of synaptotagmin membrane insertion. Thus, the current model provides an upper estimate for the magnitude of synchronous release and lower estimates for asynchronous release and short-term facilitation. We also note that in many synapses the degree of short-term facilitation is determined not only by the activation of Syt7 but also by an increase in the amplitude of $[Ca^{2+}]$ transients at the vesicular release sites during repetitive stimulation. This increase can be

attributed to various phenomena, including buffer saturation[62], action potential broadening[63], $Ca^{2+}$ channel facilitation[64] and $Ca^{2+}$ release from the intracellular stores[65,66].

For simplicity, we only modelled the activation of Syt1 C2B and Syt7 C2A domains. Considering that the C2A and C2B domains act synergistically[67], inclusion of the second C2 domains in the models will increase the overall $Ca^{2+}$/membrane affinity of synaptotagmin molecules and the $Ca^{2+}$ cooperativity. At present, the binding modalities of Syt7 to the SNARE complex remain unknown; thus, we modelled Syt7 binding to SNAREs via the tripartite interface based on structural homology[21]. However, it is worth noting that the output for the Syt1/Syt7 dual clamp model will be the same even if Syt7 binds at a different site. Furthermore, in addition to the synaptotagmins and canonical SNAREs considered in our model, different modes of $Ca^{2+}$-evoked release might utilise vesicles with distinct compositions of SNAREs and $Ca^{2+}$ sensors (e.g. VAMP4[68], VAMP7[69] and Doc2[70]).

Our default model operates on the assumption that SNAREpins, upon zippering, add energy independently of one another. However, it is possible that SNAREs on a given RRP vesicle can be coupled either mechanically[43] or via formation of supramolecular complexes, e.g. Syt1 oligomerisation[14,71,72]. In the framework of the mechanical coupling model, the partially assembled SNAREpins

interact with each other via long-range mechanical forces, mediated by the surrounding scaffolding membranes. Therefore, each clamped SNAREpin acts as an additional mechanical obstacle for the fusing membranes, thereby generating a negative feedback loop. Given that the exact energetic cost of a clamped SNAREpin remains undefined, we explored the potential influence of this mechanism by introducing an extra energy barrier of 2 $k_BT$ for each SNAREpin (Supplementary Fig. 3). This incorporation of mechanical coupling increases the number of free SNAREs required to drive fast SV fusion from approximately three to four. Consequently, this led to a substantial decrease in the SV fusion rate for [Ca$^{2+}$] below 8 μM. However, it did not significantly alter the peak release rate for more physiological [Ca$^{2+}$] at or above 8 μM.

Formation of Syt1 oligomers has been shown to strengthen the fusion clamp and inhibit spontaneous release[71,73]. Indeed, in our model we adopted an idealised representation of the fusion clamp, which is only released upon Ca$^{2+}$-activation of synaptotagmin molecules (Fig. 1). Nevertheless, it is conceivable that some of the SNAREpins might be spontaneously released from the fusion clamp, even without the Ca$^{2+}$ signal. To investigate the potential impact of this mechanism, we varied the probability that a given SNAREpin is free from fusion clamp at resting conditions (Supplementary Fig. 4). This variation resulted in an increase of spontaneous release rate but did not significantly change the evoked release in response to [Ca$^{2+}$] steps above 4–8 μM for all the clamp architectures tested.

The multifaceted interplay of mechanical coupling, high-order structural organisation of release machinery and the membrane remodelling activities of Syt1 that occur after the release of inhibition[15–21] are yet to be thoroughly explored. Nevertheless, the current model can be easily modified to implement any of the above mechanisms as well as other fusion clamp architectures. Therefore, the developed modelling framework provides a powerful and adaptable tool to complement experimental work and gain insights into how the presynaptic vesicular release machinery decodes Ca$^{2+}$ signals and translates them into complex patterns of neurotransmitter release.

## Methods

**Release of inhibition model parameters**. We considered that within the physiological range of [Ca$^{2+}$] observed at the presynaptic AZ (~50 nM–200 μM) Syt1 and Syt7 C2 domains can bind two Ca$^{2+}$ ions independently, with similar intrinsic affinities $K_d$ ~150 μM[44,45]. It has been shown that the rate of Ca$^{2+}$ binding by synaptotagmin C2 domains is limited by diffusion (~0.1–10 μM$^{-1}$ ms$^{-1}$)[48]. Therefore, we assumed $k_{on} = 1$ μM$^{-1}$ ms$^{-1}$. Because of the symmetry between the free and fully Ca$^{2+}$-bound states ($S_0$ and $S_2$, respectively, in the kinetic scheme Fig. 1b), the dissociation rate constant can be expressed as $k_{off} = k_{on} \cdot K_D$, which yielded $k_{off} = 150$ ms$^{-1}$ based on the intrinsic Ca$^{2+}$ affinity $K_d = 150$ μM. The characteristic time for synaptotagmin C2 domain rotation and membrane insertion has previously been estimated as ~10 μs[14], which corresponds to a rate of $k_{in} = 100$ ms$^{-1}$. Exponential rate constants describing the apparent rates of C2 domain dissociation from lipid membranes ($k_{diss}$) were previously measured in the presence of EGTA using stopped-flow experiments and reported to be in the ranges of 0.38–0.7 ms$^{-1}$ for Syt1 and 0.008–0.02 ms$^{-1}$ for Syt7[46–48]. We used representative values of $k_{diss} = 0.5$ ms$^{-1}$ for Syt1 and $k_{diss} = 0.015$ ms$^{-1}$ for Syt7. As demonstrated in Supplementary Note 1, the relationship between the actual rates at which C2 domain aliphatic loops dissociate from the membrane in our model ($k_{out}$) and the experimentally determined apparent rate $k_{diss}$ can be approximated as $k_{out} = k_{diss}(1 - \frac{k_{in}}{k_{diss} - 2k_{off}})$. This yields $k_{out} = 0.67$ ms$^{-1}$ for Syt1 and $k_{out} = 0.02$ ms$^{-1}$ for Syt7. The Ca$^{2+}$ and membrane binding

properties of Syt1 and Syt7 used in the model are summarised in Supplementary Table 1.

The rate of SNARE-mediated SV fusion was determined by assuming that the repulsive forces between a docked SV and the plasma membrane amount to an energy barrier of $E_0 \approx 26$ $k_BT$[42]. Overcoming this barrier requires bringing the SV to within around 1–2 nm of the plasma membrane such that membrane fusion is spontaneously induced[14]. The full assembly of a single SNARE complex from a half-zippered state has been estimated to provide $\Delta E \approx 4.5$ $k_BT$ of work towards overcoming the resting energy barrier[43]. We assumed that $\Delta E$ is made immediately available to the vesicle in the form of potential energy when a SNAREpin is freed from its synaptotagmin clamp, effectively lowering the energy barrier to membrane fusion. Thus, with $n$ uninhibited SNAREpins, the barrier to fusion has a height of $E_0 - n\Delta E$ and is spontaneously overcome through thermal fluctuations at a rate given by the Arrhenius equation $R_{rate}(n) = A \cdot \exp(-\frac{E_0 - n\Delta E}{k_BT})$ (Fig. 1c). We estimated the pre-factor $A = 2.17 \times 10^9$ s$^{-1}$, considering that a single SNARE complex can mediate fusion in vitro on a time scale of 1 s[14,43].

**Implementation of stochastic simulations**. All simulations and analysis were carried out in MATLAB 2020b, The MathWorks, Inc. The predicted responses of the release of inhibition models and the benchmark allosteric model to input [Ca$^{2+}$]($t$) traces were generated by Monte Carlo estimation from multiple stochastic simulations. Specifically, we used the direct Gillespie algorithm[49] which proceeds by iteratively generating a randomised time at which the system next changes its state and then randomly selecting the identity of the new state. The Ca$^{2+}$ binding and SV fusion dynamics of the allosteric model were described by a six-state kinetic scheme with a single occupied state which is updated at each step of the algorithm. The release of inhibition models consisted of either six (in the case of Syt1$^P$) or twelve (in the cases of Syt1$^P$/Syt1$^T$ and Syt1$^P$/Syt7$^T$) four-state kinetic schemes (Fig. 1b), one for each Syt C2 domain. Rather than updating a unique state in the resultant macroscopic Markov chain which had either $4^6$ or $4^{12}$ states, we monitored each synaptotagmin C2 domain concurrently and updated one of their states according to the algorithm. We assumed that at the start of each simulation both the Ca$^{2+}$ sensor in the benchmark allosteric model and all SNARE-associated synaptotagmin C2 domains in the release of inhibition models were in the Ca$^{2+}$ unbound state.

In simulations shown in Figs. 2–4 we did not include the mechanism for vesicle replenishment and individual stochastic simulations terminated when vesicle fusion occurred. In simulations describing vesicular release in response to a 10 × 100 Hz AP train we included a mechanism for SV replenishment. Upon vesicle fusion the release site remained unoccupied for a fixed refractory time of 2.5 ms after which a SV was replenished in the initial state with the rate of $k_{rep} = 0.02$ ms$^{-1}$, as was estimated in our previous work[28].

For each scenario, the collection of stochastic simulations yielded a set of times at which SV fusion occurred. We used the cumulative count of these vesicle fusion times, normalised to the total number of stochastic simulations, as an estimate for the expected cumulative number of vesicles exocytosed at a single release site by time $t$ ($n_T(t)$). In the absence of vesicle replenishment $n_T(t)$ corresponds to the cumulative vesicular release probability $p_v(t)$. In production, $n_T(t)$ was calculated by gathering release event times into a histogram with an adaptive bin width to capture the features of release kinetics at different temporal scales. The release rate was estimated as $\frac{dn_T(t)}{dt}$ with a

moving average smoothing applied to limit the sensitivity of peaks to stochastic variation.

**Statistics and reproducibility.** In order to assess the accuracy of these release estimates generated using the Monte Carlo approach, we compared the $n_T(t)$ values obtained from stochastic simulations of the benchmark allosteric model to the $n_T(t)$ values obtained by the numerical solution of the allosteric model's differential master equations. Given a set of $N$ release events, we found that the normalised root mean squared error in Monte Carlo predictions converged at a rate of approximately $\frac{1}{\sqrt{N}}$ (Supplementary Fig. 5). For each scenario considered with both the allosteric and release of inhibition models, by default, stochastic simulations were set to continue until $10^5$ release events had been recorded. This corresponded to an average error of less than 0.1%. However, due to resource constraints and differences in computational demand between scenarios, this was not always achieved, and the total number of simulated release events varied. Across all scenarios, the minimum number of release events recorded was 2250 (Syt1$^P$/Syt1$^T$ at a 2 μM step), corresponding to an average error of less than 1%, making this the expected upper limit of prediction errors due to stochastic variation.

**Simulation of [Ca$^{2+}$] transients within a presynaptic terminal.** Three-dimensional modelling of AP-evoked presynaptic Ca$^{2+}$ influx, buffering, diffusion and extrusion was performed in the Virtual Cell (VCell) simulation environment (vcell.org) as described in detail in our previous studies. Specifically, [Ca$^{2+}$]($t$) transients approximating Ca$^{2+}$ dynamics at the AZ of small excitatory boutons (Figs. 3 and 4) and MFB terminals (Figs. 5 and 6) were computed using the VCell models described in ref. 29 and ref. 28 respectively. The properties of endogenous Ca$^{2+}$ buffers used in the model were previously estimated in refs. 74–80. and are shown in Supplementary Table 2.

**Reporting summary.** Further information on research design is available in the Nature Portfolio Reporting Summary linked to this article.

## Data availability

The data for the figures is provided with the code and any remaining information can be obtained on request from the corresponding authors.

## Code availability

Custom MATLAB codes are provided with the paper within Norman_et_al_code.zip or can be accessed at this GitHub repository.

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

## Acknowledgements

We are grateful to Drs Dimitri Kullmann, James Rothman and Katalin Toth for reading the manuscript and providing critical feedback. This work was supported by UKRI MRC Project Grant MR/T002786/1 (Y.T. and K.E.V.), UKRI BBSRC/NC3R Project Grant NC/X002233/1 (K.E.V.), National Institute of Health (NIH) grant NS133091 (S.S.K. and K.E.V.) and EPSRC Mathematics for Real-World Systems II Centre for Doctoral Training EP/S007016/1 (C.A.N. and Y.T.). The Virtual Cell is supported by NIH Grant R24 GM137787 from the National Institute for General Medical Sciences.

## Author contributions

Design and conceptualisation: C.A.N., S.S.K., Y.T. and K.E.V. Constraining model parameters: C.A.N., S.S.K. and K.E.V. Implementation of the computational model: C.A.N. and Y.T. Performing simulations: C.A.N. Writing paper, review and editing C.A.N., S.S.K., Y.T. and K.E.V.

## Competing interests

The authors declare no competing interests.
