## [Peer Review File · Communications Biology]

Reviewers' comments:

Reviewer #1 (Remarks to the Author):

Norman and colleagues presented an experimentally constrained computational model in their manuscript, which quantitatively evaluated the impact of the fusion clamp's molecular architecture on SV exocytosis. The authors discovered that the "release-of-inhibition" model can account for rapid calcium-activated SV fusion, and that the dual binding of Syt1 and Syt7 to the same SNARE complex enables synergistic regulation of neurotransmitter release's kinetics and plasticity. Although the results are robust and of high quality, certain interpretations require improvement. The authors should address the comments provided to make this study a strong contender for Communications Biology.

According to prior research, the authors assumed that each RRP vesicle was linked to six SNARE complexes. However, based on their findings in this manuscript, the authors predicted that at least three uninhibited SNARE complexes were necessary for fast fusion. To avoid misunderstandings, the authors should elaborate on this point.

The study also assumed that each SNAREpin contributed independently to synaptic vesicle fusion. Is it different if SNARE complexes and other components form supercomplexes such as ring-like oligomers?

Reviewer #2 (Remarks to the Author):

In this manuscript, Norman et al. present a numerical simulation study of detailed biophysical models for calcium-activated synaptic vesicle fusion mediated by SNARE and synaptotagmin proteins (Syt). In particular, they explore the potential contribution of Syt1 and Syt7 to the "release of inhibition" hypothesis. With their models calibrated using published experimental data from the previous literature, they explore the time scales for calcium-activated vesicle fusion and the condition under which these agree with generic experimental observations, in particular regarding short-term plasticity. The authors conclude that their numerical explorations speak in favor of the "release of inhibition" mechanism.

Overall, I find the manuscript well written. I think the authors should be congratulated for their neat modelling and the strong attention given to the biochemical context. I believe computational models are especially useful when the main dynamical processes elude experimental observation (which is the case here because of the small spatial scales involved) and when the model is developed with a very good knowledge of the experimental data of the field (which is also the case here). However, I

think that, in its current state, the manuscript is still very much an academic exercise and that there is room for improvement of its potential impact in the field. Moreover a couple of modelling details are to be more clearly described.

MAJOR POINTS

1) I was a bit disappointed by the level of model-experiments interactions in the article. I agree that the development of the model is rooted in data from the experimental literature - regarding the molecular mechanisms included and the calibration of the parameters. However, beyond model development, I am sorry to say that the feedback from the simulation results to the experimental data left me hungry for more. Figure 2 to Figure 6 present simulation results that emulate concrete experimental situations but it is not clear how these simulations compare with experimental data. The manuscript proposes only a few comparison points with experimental results but these comparisons are rare and usually not of a quantitative nature. For instance how do experimental quantifications compare with the release rates shown in figure 2A or 3BC, exactly (ie where do the experimental measurements position on the figures)? Likewise, how do experimental quantifications of short-term plasticity (paired-pulse ratios) compare with the curves of Figure 4C, exactly?

2) Regarding your spatially-explicit model (VCell), I see no diffusion coefficient for free calcium in Supplementary Table 1. Does that mean that free calcium is not allowed to diffuse in your model? But can only bind / unbind locally to a buffer, and diffuse only when bound? Of course, the diffusion coefficient of the buffers themselves are much slower than free calcium (20 - 85 micrometer²/s compared to > 200) and that must strongly facilitate the formation of a local nanodomain around the VGCC cluster. However, what is the impact of this modelling choice on the match between experimental and simulations for local calcium dynamics? Can you provide data that would support the fact that the spatio-temporal dynamics of your calcium traces are indeed realistic? And what exact species binds to Syt1 and Syt7 in the model? I don't suppose it is free Ca if free Ca does not diffuse, and thus cannot reach Syt targets except if unbinding from a buffer happens exactly in the very vicinity of a Syt molecule.

*MINOR POINTS

1) The relation between k_{diss} and (k_{out}, k_{in}) is not clear at all in the article. First, I may be mistaken but I see nowhere a definition of what exact reaction k_{diss} is the rate of? I do not see any k_{diss} -dependent reaction in the kinetic schemes shown (Fig 1B and sup fig 2A). The equation relating k_{diss} , k_{out} and k_{in} in the caption of supp figure 2 is not justified and is not sufficient for me to understand what is k_{diss} for.

2) I did not understand what was the rationale for the comparison between your model simulation and that of the "allosteric model". What happens if your simulations are close to that of the allosteric model? Is it good or bad? Why? Does it tell anything on the comparison between your model and experimental data?

3) As nicely shown in Supp Figure 3, the results you show in your figure are averaged over a number of realisations of the stochastic simulations that is so large that your results are probably perfectly converged to the integration of the corresponding mean field, deterministic ODEs. So why do you waste computer time using stochastic simulations instead of numerical integration of the mean-field ODEs (at least for non-spatially explicit simulations)?

4) Recent simulation studies have suggested that the contribution of the presynaptic endoplasmic reticulum to PPR (at least at CA3-CA1 synapses), see e.g., Singh et al, *Commun Biol*, 2021 4:241. How does this line of results impact your main conclusions? A priori the fact that simulations that do not account for the presence of the ER provide a realistic description the local dynamics contradicts the idea that the ER is crucial for PPR or vesicle fusion. I think this point should be tackled, at least in the discussion section.

Reviewer #3 (Remarks to the Author):

In this work, a computational model is presented for neurotransmitter release at synapses. The model is tested by comparison to an allosteric model of release at the calyx of Held giant synapse. The computational model consists of inhibited states of a system that are released upon calcium binding. Two different inhibited states are considered based on the available structural and functional data about SNARE complex interactions with synaptotagmins and complexins. It is assumed that the primary interface between Syt1 (Syt2/Syt9) and the SNARE complex is always present at resting calcium concentration, while the tripartite interface is optional and could be occupied by either Syt1 or Syt7. The computational model does not depend on the particular atomic resolution structures, so it could in principle also apply to other interactions between Syt7 with the SNARE complex – note that a structure of Syt7 bound to the SNARE complex has not yet been solved. The model presented here is quite general and it is agnostic to the actual structural arrangements of complexes between the RRP vesicle and the plasma membrane. Overall, this computational model provides an elegant and informative framework for describing synaptic neurotransmitter release.

Comments:

1. The model assumes that there is a total of six pre-fusion inhibited SNAREpins associated with each RRP vesicle based on previously published analysis of cryo-electron tomography data (reference 34). However, considering that such hexameric arrangements seem unusual, does their model critically depend on this number? I suspect not, considering that the inhibited SNAREpins presumably do not hinder triggered fusion (see also the next comment).

2. Figure 2D suggests that 3 unclamped SNAREpins optimally describe the calyx of Held data. Does this imply that the 3 remaining SNAREpins are still clamped, i.e., there is no penalty for a clamped SNAREpin in the model?

3. Please clarify the difference in modelling parameters between Syt1 and Syt7. In the Methods, it is stated that both have similar calcium binding affinities and on-rates. It is also assumed that the rates of dissociation from membranes are different for Syt1 and Syt7. Is that the only difference between Syt1 and Syt7 that is assumed in the model? It would be helpful if all the rate constants and parameters could be summarized in a table.

4. Please clarify the model for the paired-pulse (synaptic plasticity) experiments. Is it assumed that the same vesicle is involved in both release processes, e.g., akin a “kiss-and-run” mechanism?

Response to the reviewers

Norman et al, Communications Biology Manuscript COMMSBIO-23-1227-T

We are grateful to the reviewers for careful consideration of the manuscript and their valuable comments. We have taken on board all of their specific suggestions, and have revised the paper. The revised manuscript contains 3 new supplementary figures (Supplementary Figs. 2 – 4) and one new Supplementary Table 2. We have also revised the text to address the reviewers' questions.

The point-by-point response to each reviewer is provided below.

Reviewers' comments are italicised.

Our responses (as well as changes in the revised manuscript) are in blue colour.

For easy reference, queries from all three reviewers were renumbered throughout the response.

Reviewer 1

Q1

According to prior research, the authors assumed that each RRP vesicle was linked to six SNARE complexes. However, based on their findings in this manuscript, the authors predicted that at least three uninhibited SNARE complexes were necessary for fast fusion. To avoid misunderstandings, the authors should elaborate on this point.

We followed the reviewer's suggestion and provided clarification in the text that '...once three out of six SNAREpins are simultaneously unclamped, vesicle fusion is inevitable...'. In addition, we now included new simulations showing that vesicular fusion takes place when at least three SNAREpins were free from the fusion clamp, regardless of the total number of SNAREpins on a given vesicle (Supplementary Figure 2).

Q2

The study also assumed that each SNAREpin contributed independently to synaptic vesicle fusion. Is it different if SNARE complexes and other components form supercomplexes such as ring-like oligomers?

In the revised manuscript we considered the possible effect of cooperative behaviour of SNAREs and included the following text and results (Supplementary Figures 3 and 4):

'...Our default model operates on the assumption that SNAREpins, upon zippering, add energy independently of one another. However, it is possible that SNAREs on a given RRP vesicle can be coupled either mechanically (Manca et al., 2019) or via formation of supramolecular complexes, e.g. Syt1 oligomerisation (Rothman et al., 2017; Tagliatti et al., 2020; Wang et al., 2014). In the framework of the mechanical coupling model, the partially assembled SNAREpins interact with each other via long-range mechanical forces, mediated by the surrounding scaffolding membranes. Therefore, each clamped SNAREpin acts as an additional mechanical obstacle for the fusing membranes, thereby generating a negative feedback loop. Given that the exact energetic cost of a clamped SNAREpin remains undefined, we explored the potential influence of this mechanism by introducing an extra energy barrier of 2 $k_B T$ for each SNAREpin (Supplementary Figure 3). This incorporation of mechanical coupling increases the number of free SNAREs required to drive fast SV fusion from approximately three to four. Consequently, this led to a substantial decrease in the SV fusion rate for $[Ca^{2+}]$ concentrations below 8 μM . However, it did not significantly alter the peak release rate for physiological $[Ca^{2+}]$ concentrations at or above 8 μM .

Formation of Syt1 oligomers has been shown to strengthen the fusion clamp and inhibit the spontaneous release (Courtney et al., 2021; Tagliatti et al., 2019). Indeed, in our model, we adopted an idealised representation of the fusion clamp, which is only released upon Ca^{2+} activation of synaptotagmin molecules (Figure 1). Nevertheless, it is conceivable that some of the SNAREpins might be spontaneously released from the fusion clamp, even without the Ca^{2+} signal. To investigate the potential impact of this mechanism, we varied the probability that a given SNAREpin is free from fusion clamp at resting conditions (Supplementary Fig. 4). This variation resulted in the increase of spontaneous release rate but did not significantly

change the evoked release in response to $[Ca^{2+}]$ steps above 4 - 8 μM for all the clamp architectures tested.

The multifaceted interplay of mechanical coupling, high-order structural organisation of release machinery and the membrane remodelling activities of Syt1 that occur after the release of inhibition (Arac et al., 2006; Hui et al., 2009; Ma et al., 2017; Martens et al., 2007; van den Bogaart et al., 2011; Wu and Schulten, 2014; Zhou et al., 2017) are yet to be thoroughly explored...'

Reviewer 2

Q3

...I was a bit disappointed by the level of model-experiments interactions in the article. I agree that the development of the model is rooted in data from the experimental literature - regarding the molecular mechanisms included and the calibration of the parameters. However, beyond model development, I am sorry to say that the feedback from the simulation results to the experimental data left me hungry for more. Figure 2 to Figure 6 present simulation results that emulate concrete experimental situations but it is not clear how these simulations compare with experimental data. The manuscript proposes only a few comparison points with experimental results but these comparisons are rare and usually not of a quantitative nature. For instance how do experimental quantifications compare with the release rates shown in figure 2A or 3BC, exactly (ie where do the experimental measurements position on the figures)? Likewise, how do experimental quantifications of short-term plasticity (paired-pulse ratios) compare with the curves of Figure 4C, exactly?...

...I did not understand what was the rationale for the comparison between your model simulation and that of the "allosteric model". What happens if your simulations are close to that of the allosteric model? Is it good or bad? Why? Does it tell anything on the comparison between your model and experimental data?...

We agree with the reviewer that it is important to relate the results of our simulations to the experimental data recorded in live synapses. Whilst the vesicular release has been measured with millisecond precision in many synapses, the shape of $[Ca^{2+}]$ transients in the vicinity of RRP vesicles could not be directly measured due to technical limitations. Thus, the direct comparison of the model output to experimental data is not feasible.

To overcome this limitation, we used the following approach.

- (i) As a model input, we approximated Ca^{2+} transients at vesicular release sites using experimentally constrained models of presynaptic Ca^{2+} dynamics that we have previously developed in VCell simulation environment. Indeed, we have demonstrated that these models could well describe Ca^{2+} dynamics in different types of synapses, including small central glutamatergic synapses and hippocampal mossy-fibre boutons (Chamberland et al., 2018; Chamberland et al., 2020; Ermolyuk et al., 2013; Mendonca et al., 2022; Timofeeva and Volynski, 2015).
- (ii) We used the allosteric model developed by Lou et al. as a benchmark for the experimental data (Lou et al., 2005). This empirical model has been shown to quantitatively describe the Ca^{2+} activation of vesicular release in many functionally and structurally distinct central synapses, including calyx of Held, GABAergic terminals of hippocampal parvalbumin-containing basket cells (Bucurenciu et al.,

2008), glutamatergic terminals of hippocampal granular (Chamberland et al., 2018; Vyleta and Jonas, 2014) and CA3 (Ermolyuk et al., 2013) pyramidal neurons.

We included the predictions of the allosteric model in Figures 2 – 6 as a quantitative approximation of how a modelled RRP vesicle would respond to different $[Ca^{2+}]$ transients in live synapses. We have included this reasoning in the revised manuscript. As noted in the manuscript, the allosteric model does not account for the dynamic changes in the state of vesicular release machinery caused by repetitive stimulation and therefore cannot adequately describe short-term synaptic facilitation. As such, in Figures 5 and 6 we also directly compared the results of our simulations to the available experimental data from hippocampal MFB terminals.

Q4

Regarding your spatially-explicit model (VCell), I see no diffusion coefficient for free calcium in Supplementary Table 1. Does that mean that free calcium is not allowed to diffuse in your model? But can only bind / unbind locally to a buffer, and diffuse only when bound? Of course, the diffusion coefficient of the buffers themselves are much slower than free calcium (20 - 85 micrometer²/s compared to > 200) and that must strongly facilitate the formation of a local nanodomain around the VGCC cluster. However, what is the impact of this modelling choice on the match between experimental and simulations for local calcium dynamics?

We are grateful to the reviewer for highlighting this omission. Indeed, in our VCell model of presynaptic Ca^{2+} dynamics Ca^{2+} is free to diffuse with the diffusion coefficient $D_{Ca^{2+}} = 220 \mu m^2 s^{-1}$ (see (Goswami et al., 2012; Meinrenken et al., 2002)). This value has been added to Supplementary Table 2.

Can you provide data that would support the fact that the spatio-temporal dynamics of your calcium traces are indeed realistic? And what exact species binds to Syt1 and Syt7 in the model? I don't suppose it is free Ca if free Ca does not diffuse, and thus cannot reach Syt targets except if unbinding from a buffer happens exactly in the the very vicinity of a Syt molecule

The parameters of the VCell model have been experimentally constrained in our previous work and validated by comparing the computed presynaptic Ca^{2+} dynamics to the experimental AP-evoked Ca^{2+} fluorescence transients recored in small excitatory boutons (Ermolyuk et al., 2013; Mendonca et al., 2022; Timofeeva and Volynski, 2015) or MFB terminals (Chamberland et al., 2018; Chamberland et al., 2020)).

Q5

The relation between k_{diss} and (k_{out} , k_{in}) is not clear at all in the article. First, I may be mistaken but I see nowhere a definition of what exact reaction k_{diss} is the rate of? I do not see any k_{diss} -dependent reaction in the kinetic schemes shown (Fig 1B and sup fig 2A). The equation relating k_{diss} , k_{out} and k_{in} in the caption of supp figure 2 is not justified and is not sufficient for me to understand what is k_{diss} for.

The apparent dissociation constant (k_{diss}) describes the exponential rates at which Syt1 and Syt7 C2 domains dissociate from lipid membranes when Ca^{2+} is rapidly removed using dilution in EGTA buffers in stopped-flow experiments (Brandt et al., 2012; Davis et al., 1999; Hui et al., 2005). We used representative values of $k_{diss} = 0.5 \text{ ms}^{-1}$ for Syt1 (0.38 - 0.7 ms^{-1} reported range) and $k_{diss} = 0.015 \text{ ms}^{-1}$ for Syt7 (0.008 - 0.02 ms^{-1} reported range).

As demonstrated in Supplementary Appendix the relationship between the actual rates at which C2 domain aliphatic loops dissociate from the membrane in our model (k_{out}) and the experimentally determined apparent rate k_{diss} can be approximated as

$$k_{out} = k_{diss} \left(1 - \frac{k_{in}}{k_{diss} - 2k_{off}} \right). \text{ This yields } k_{out} = 0.67 \text{ ms}^{-1} \text{ for Syt1 and } k_{out} = 0.02 \text{ ms}^{-1} \text{ for Syt7.}$$

We have highlighted this in the method section of the revised manuscript.

Q6

As nicely shown in Supp Figure 3, the results you show in your figure are averaged over a number of realisations of the stochastic simulations that is so large that your results are probably perfectly converged to the integration of the corresponding mean field, deterministic ODEs. So why do you waste computer time using stochastic simulations instead of numerical integration of the mean-field ODEs (at least for non-spatially explicit simulations)?

There are two key reasons why a Monte Carlo approach was used over a deterministic one for simulating the release of inhibition models' responses to Ca^{2+} stimuli:

(i) These models explicitly contain discrete events which cannot be accounted for with a solely continuous model. For example, when a SNARE is released from inhibition by all its associated synaptotagmins, the rate of SV fusion increases instantaneously. The Monte Carlo approach allows us to approximate the continuous model elements while accounting for these discrete step changes.

(ii) It is typical for vesicle replenishment to be modelled with a fixed refractory period (2.5 ms in Figures 5 and 6). This requires explicit memorisation of the time the previous event (fusion) occurred, violating the Markov property. The resulting models fall into the category of semi-Markov chains which are more challenging to simulate deterministically but can be straightforwardly accommodated in stochastic simulations.

Q7

Recent simulation studies have suggested that the contribution of the presynaptic endoplasmic reticulum to PPR (at least at CA3-CA1 synapses), see e.g., Singh et al, Commun Biol, 2021 4:241. How does this line of results impact your main conclusions? A priori the fact that simulations that do not account for the presence of the ER provide a realistic description the local dynamics contradicts the idea that the ER is crucial for PPR or vesicle fusion. I think this point should be tackled, at least in the discussion section.

As suggested by the reviewer we have included the following clarification in discussion section: '...We also note that in many synapses the degree of short-term facilitation is determined not only by the activation of Syt7 but also by an increase in the amplitude of $[Ca^{2+}]$

transients at the vesicular release sites during repetitive stimulation. This increase can be attributed to various phenomena, including buffer saturation (Matveev et al., 2004), action potential broadening (Jackson et al., 1991), Ca^{2+} channel facilitation (Mochida et al., 2008) and Ca^{2+} release from the intracellular stores (Emptage et al., 2001; Singh et al., 2021)...'

Reviewer 3

Q8

The model assumes that there is a total of six prefusion inhibited SNAREpins associated with each RRP vesicle based on previously published analysis of cryo-electron tomography data (reference 34). However, considering that such hexameric arrangements seem unusual, does their model critically depend on this number? I suspect not, considering that the inhibited SNAREpins presumably do not hinder triggered fusion (see also the next comment).

We agree with the reviewer, that the exact number of SNARE complexes in each RRP vesicle is still undetermined. Therefore, we explored the effect of varying the number of SNAREpins per vesicle (specifically four, six and eight). We found that for all clamp architectures vesicular fusion took place when at least three SNAREpins were free from the fusion clamp, regardless of the total number of SNAREpins on a given vesicle. However, we observed a correlation between the peak fusion rate and the number of available SNAREpins. For every $[\text{Ca}^{2+}]$ tested, the peak release rate was at its highest with eight SNAREpins per vesicle and at its lowest with four. This pattern could be explained by the fact that the state with three unclamped SNAREpins was reached more quickly when there were more total SNAREpins on the vesicle. This set of data is shown in Supplementary Figure 2 and described in the Results section of the revised manuscript.

Q9

Figure 2D suggests that 3 unclamped SNAREpins optimally describe the calyx of Held data. Does this imply that the 3 remaining SNAREpins are still clamped, i.e., there is no penalty for a clamped SNAREpin in the model?

In the revised manuscript we considered the possible effect of cooperative behaviour of SNAREs and included additional text and results (Supplementary Figures 3 and 4). Please see the detailed response to Q2 above.

Q10

Please clarify the difference in modelling parameters between Syt1 and Syt7. In the Methods, it is stated that both have similar calcium binding affinities and on-rates. It is also assumed that the rates of dissociation from membranes are different for Syt1 and Syt7. Is that the only difference between Syt1 and Syt7 that is assumed in the model? It would be helpful if all the rate constants and parameters could be summarized in a table.

Yes, based on the available literature we assumed that the higher Ca^{2+} /membrane affinity of Syt7 is primarily due to its slower dissociation kinetics from membranes. Hence, we used the

same k_{on} , k_{off} and k_{in} but different k_{out} rates for Syt1 and Syt7. As suggested by the reviewer we have summarised these values in Supplementary Table 1.

Q11

Please clarify the model for the paired-pulse (synaptic plasticity) experiments. Is it assumed that the same vesicle is involved in both release processes, e.g., akin a “kiss-and-run” mechanism?

In Figure 4 we specifically aimed to test how the molecular architecture of the synaptotagmin clamp under a given RRP vesicles shapes its probability of fusion during paired-pulse stimulation. Therefore, in this set of simulations we chose not to model SV replenishment. Specifically, if a vesicle was released at the first AP, there would be no release at the second AP. This approach has been clarified in the manuscript, in accordance with the reviewer's suggestion. We would also like to point out that, in the current model, a given SV will either fuse entirely or not at all. Consequently, our model does not accommodate the possibility of a 'kiss-and-run' event, a mechanism that might be considered in future investigations.

New figures

Supplementary Figure 2. Effect of total number of SNAREpins on Ca^{2+} activation of SV fusion.

(A) Top, time-course of vesicular release rate simulated in response to a $16 \mu\text{M}$ $[\text{Ca}^{2+}]$ step for the single and dual synaptotagmin/SNARE clamp architectures considered in the model with four, six, or eight total SNAREpins per vesicle (solid coloured traces) and for the benchmark allosteric model (dashed grey trace). Bottom, time evolution of the mean number of unclamped SNAREpins ('Free SNAREs') on all docked SVs (solid lines), and on SVs at the instance of fusion (dotted lines) in response to the $16 \mu\text{M}$ $[\text{Ca}^{2+}]$ step. Shaded area indicates 1 standard deviation each side of the mean. Each time point includes data from a 0.15 ms bin.

(B) Dependency of the peak release rate (achieved within 10 ms) on the amplitude of the $[\text{Ca}^{2+}]$ step. For each $[\text{Ca}^{2+}]$ step and fusion clamp architecture at least $N = 100,000$ stochastic simulations were performed with at least 1,000 vesicular fusion events recorded during the first 10 ms time window.

Supplementary Figure 3. Effect of mechanical coupling among SNAREpins on the same vesicle on Ca^{2+} activation of SV fusion.

To model mechanical coupling, we assumed that each clamped SNAREpin acts as an additional mechanical obstacle for the fusing membranes, thereby generating a negative feedback loop by introducing an extra energy barrier of $2 k_B T$.

(A) Top, time-course of vesicular release rate simulated in response to a $16 \mu M [Ca^{2+}]$ step for the single and dual synaptotagmin/SNARE clamp architectures considered in the model with each unclamped SNAREpin contributing either $0 k_B T$ or $2 k_B T$ to the total energy barrier (solid coloured traces) and for the benchmark allosteric model (dashed grey trace). Bottom, time evolution of the mean number of unclamped SNAREpins ('Free SNAREs') on all docked SVs (solid lines), and on SVs at the instance of fusion (dotted lines) in response to the $16 \mu M [Ca^{2+}]$ step. Shaded area indicates 1 standard deviation each side of the mean. Each time point includes data from a $0.15 ms$ bin.

(B) Dependency of the peak release rate (achieved within 10 ms) on the amplitude of the $[Ca^{2+}]$ step. For each $[Ca^{2+}]$ step and fusion clamp architecture at least $N = 100,000$ stochastic simulations were performed with at least 1,000 vesicular fusion events recorded during the first 10 ms time window. For clarity, only points with release rate $> 10^{-6} ms^{-1}$ are shown.

Supplementary Figure 4. Modelling of spontaneous release of fusion clamp

Dependency of the peak release rate (achieved within 20 ms) on the amplitude of the [Ca²⁺] step for a range of cases in which the probability that each individual SNAREpin was initially unclamped varied between 0% and 2%, as indicated. For each [Ca²⁺] step at least $N = 200,000$ stochastic simulations were performed with at least 2,000 vesicular fusion events recorded.

Reference List

- Arac,D., Chen,X., Khant,H.A., Ubach,J., Ludtke,S.J., Kikkawa,M., Johnson,A.E., Chiu,W., Sudhof,T.C., and Rizo,J. (2006). Close membrane-membrane proximity induced by Ca(2+)-dependent multivalent binding of synaptotagmin-1 to phospholipids. *Nat. Struct. Mol. Biol.* **13**, 209-217.
- Brandt,D.S., Coffman,M.D., Falke,J.J., and Knight,J.D. (2012). Hydrophobic contributions to the membrane docking of synaptotagmin 7 C2A domain: mechanistic contrast between isoforms 1 and 7. *Biochemistry* **51**, 7654-7664.
- Bucurenciu,I., Kulik,A., Schwaller,B., Frotscher,M., and Jonas,P. (2008). Nanodomain coupling between Ca²⁺ channels and Ca²⁺ sensors promotes fast and efficient transmitter release at a cortical GABAergic synapse. *Neuron*. **57**, 536-545.
- Chamberland,S., Timofeeva,Y., Evstratova,A., Norman,C.A., Volynski,K., and Tã³th,K. (2020). Slow-decaying presynaptic calcium dynamics gate long-lasting asynchronous release at the hippocampal mossy fiber to CA3 pyramidal cell synapse. *Synapse*. **74**, e22178.
- Chamberland,S., Timofeeva,Y., Evstratova,A., Volynski,K., and Toth,K. (2018). Action potential counting at giant mossy fiber terminals gates information transfer in the hippocampus. *Proc. Natl. Acad. Sci. U. S. A.* **115**, 7434-7439.
- Courtney,K.C., Vevea,J.D., Li,Y., Wu,Z., Zhang,Z., and Chapman,E.R. (2021). Synaptotagmin 1 oligomerization via the juxtamembrane linker regulates spontaneous and evoked neurotransmitter release. *Proc. Natl. Acad. Sci. U. S. A* **118**.
- Davis,A.F., Bai,J., Fasshauer,D., Wolowick,M.J., Lewis,J.L., and Chapman,E.R. (1999). Kinetics of synaptotagmin responses to Ca²⁺ and assembly with the core SNARE complex onto membranes. *Neuron*. **24**, 363-376.
- Emptage,N.J., Reid,C.A., and Fine,A. (2001). Calcium stores in hippocampal synaptic boutons mediate short-term plasticity, store-operated Ca²⁺ entry, and spontaneous transmitter release. *Neuron*. **29**, 197-208.
- Ermolyuk,Y.S., Alder,F.G., Surges,R., Pavlov,I.Y., Timofeeva,Y., Kullmann,D.M., and Volynski,K.E. (2013). Differential triggering of spontaneous glutamate release by P/Q-, N- and R-type Ca(2+) channels. *Nat. Neurosci.* **16**, 1754-1763.
- Goswami,S.P., Bucurenciu,I., and Jonas,P. (2012). Miniature IPSCs in Hippocampal Granule Cells Are Triggered by Voltage-Gated Ca²⁺ Channels via Microdomain Coupling. *J. Neurosci.* **32**, 14294-14304.
- Hui,E., Bai,J., Wang,P., Sugimori,M., Llinas,R.R., and Chapman,E.R. (2005). Three distinct kinetic groupings of the synaptotagmin family: candidate sensors for rapid and delayed exocytosis. *Proc. Natl. Acad. Sci. U. S. A.* **102**, 5210-5214.
- Hui,E., Johnson,C.P., Yao,J., Dunning,F.M., and Chapman,E.R. (2009). Synaptotagmin-mediated bending of the target membrane is a critical step in Ca(2+)-regulated fusion. *Cell*. **138**, 709-721.
- Jackson,M.B., Konnerth,A., and Augustine,G.J. (1991). Action potential broadening and frequency-dependent facilitation of calcium signals in pituitary nerve terminals. *Proc. Natl. Acad. Sci. U. S. A* **88**, 380-384.
- Lou,X., Scheuss,V., and Schneggenburger,R. (2005). Allosteric modulation of the presynaptic Ca²⁺ sensor for vesicle fusion. *Nature*. **435**, 497-501.
- Ma,L., Cai,Y., Li,Y., Jiao,J., Wu,Z., O'Shaughnessy,B., De,C.P., Karatekin,E., and Zhang,Y. (2017). Single-molecule force spectroscopy of protein-membrane interactions. *Elife*. **6**. pii: e30493. doi: 10.7554/eLife.30493., e30493.
- Manca,F., Pincet,F., Truskinovsky,L., Rothman,J.E., Foret,L., and Caruel,M. (2019). SNARE machinery is optimized for ultrafast fusion. *Proc. Natl. Acad. Sci. U. S. A* **116**, 2435-2442.
- Martens,S., Kozlov,M.M., and McMahon,H.T. (2007). How synaptotagmin promotes membrane fusion. *Science*. **316**, 1205-1208.
- Matveev,V., Zucker,R.S., and Sherman,A. (2004). Facilitation through buffer saturation: constraints on endogenous buffering properties. *Biophys. J.* **86**, 2691-2709.

Meinrenken,C.J., Borst,J.G., and Sakmann,B. (2002). Calcium secretion coupling at calyx of held governed by nonuniform channel-vesicle topography. *J. Neurosci.* 22, 1648-1667.

Mendonca,P.R.F., Tagliatti,E., Langley,H., Kotzadimitriou,D., Zamora-Chimal,C.G., Timofeeva,Y., and Volynski,K.E. (2022). Asynchronous glutamate release is enhanced in low release efficacy synapses and dispersed across the active zone. *Nat. Commun.* 13, 3497.

Mochida,S., Few,A.P., Scheuer,T., and Catterall,W.A. (2008). Regulation of presynaptic $ca(v)2.1$ channels by $ca(2+)$ sensor proteins mediates short-term synaptic plasticity. *Neuron.* 57, 210-216.

Rothman,J.E., Krishnakumar,S.S., Grushin,K., and Pincet,F. (2017). Hypothesis - buttressed rings assemble, clamp, and release SNAREpins for synaptic transmission. *FEBS Lett.* 591, 3459-3480.

Singh,N., Bartol,T., Levine,H., Sejnowski,T., and Nadkarni,S. (2021). Presynaptic endoplasmic reticulum regulates short-term plasticity in hippocampal synapses. *Commun. Biol.* 4, 241.

Tagliatti,E., Bello,O.D., Mendonca,P.R.F., Kotzadimitriou,D., Nicholson,E., Coleman,J., Timofeeva,Y., Rothman,J.E., Krishnakumar,S.S., and Volynski,K.E. (2020). Synaptotagmin 1 oligomers clamp and regulate different modes of neurotransmitter release. *Proc. Natl. Acad. Sci. U. S. A.* 117, 3819-3827.

Tagliatti,E., Bello,O.D., Mendonca,P.R.F., Kotzadimitriou,D., Nicholson,E., Coleman,J., Timofeeva,Y., Rothman,J.E., Krishnakumar,S.S., and Volynski,K.E. (2019). Synaptotagmin 1 oligomers clamp and regulate different modes of neurotransmitter release. *BioRxiv* 594051.

Timofeeva,Y., and Volynski,K.E. (2015). Calmodulin as a major calcium buffer shaping vesicular release and short-term synaptic plasticity: facilitation through buffer dislocation. *Front Cell Neurosci.* 9:239. doi: 10.3389/fncel.2015.00239. eCollection@2015., 239.

van den Bogaart,G., Thutupalli,S., Risselada,J.H., Meyenberg,K., Holt,M., Riedel,D., Diederichsen,U., Herminghaus,S., Grubmüller,H., and Jahn,R. (2011). Synaptotagmin-1 may be a distance regulator acting upstream of SNARE nucleation. *Nat. Struct. Mol. Biol.* 18, 805-812.

Vyleta,N.P., and Jonas,P. (2014). Loose coupling between Ca^{2+} channels and release sensors at a plastic hippocampal synapse. *Science.* 343, 665-670.

Wang,J., Bello,O., Auclair,S.M., Wang,J., Coleman,J., Pincet,F., Krishnakumar,S.S., Sindelar,C.V., and Rothman,J.E. (2014). Calcium sensitive ring-like oligomers formed by synaptotagmin. *Proc. Natl. Acad. Sci. U. S. A.* 201415849.

Wu,Z., and Schulten,K. (2014). Synaptotagmin's role in neurotransmitter release likely involves $Ca(2+)$ -induced conformational transition. *Biophys. J.* 107, 1156-1166.

Zhou,Q., Zhou,P., Wang,A.L., Wu,D., Zhao,M., Sudhof,T.C., and Brunger,A.T. (2017). The primed SNARE-complexin-synaptotagmin complex for neuronal exocytosis. *Nature.* 548, 420-425.

REVIEWERS' COMMENTS:

Reviewer #1 (Remarks to the Author):

I am content with the revised manuscript. The authors' response addressed all of my inquiries thoroughly.

Reviewer #2 (Remarks to the Author):

The authors have satisfactorily addressed my concerns in the revised version. Therefore, I have no further comments.

Reviewer #3 (Remarks to the Author):

I would like to thank the authors for addressing my questions. I have no further comments and fully support acceptance of this revised manuscript.